# OpenReview forum: "Signature-Guided Adversarial Attacks On Healthcare LLMs: Exposing PII Leakage In RAG Systems"
_ICLR.cc/2026/Conference — Submitted to ICLR 2026_

### Official Review · Reviewer_yn4e · 2025-10-24

**Soundness:** 2
**Presentation:** 2
**Contribution:** 2
**Rating:** 2
**Confidence:** 4

**Summary:**

The paper proposes an attack framework against LLM healthcare assistants capable of performing RAG over medical notes to extract PII of patients. Despite de-identification methods which must be employed to ensure legal privacy compliance, information leakage remains a risk, with past works attempting to protect against it through model fine-tuning and prompting. To test the vulnerability of such methods, they construct a healthcare RAG pipeline with access to a dataset of medical notes with synthetic PII, and use an RLHF model (GPT-4) with a safety-oriented system prompt. They propose an adversary with access to de-identified medical notes which is capable of prompting this victim model with the intent of identifying patients/leaking sensitive patient data. As the target data contains many entries, the paper proposes a method for creating a "unique signature" of the target notes they wish to identify by selecting the most unique (distinct from other medical notes) medical terms and corresponding contexts for a given patient. This signature is then used as part of an iterative adversarial prompting strategy to extract identifiable information, such as names, from the victim RAG model.

**Strengths:**

- The topic that the paper addresses is important.
- The method is motivated and demonstrates stronger performance than other methods.

**Weaknesses:**

- The proposal reads like a stack of existing tools and ideas that are used to solve a real world problem rather than a novel method published at a scientific conference or journal.
- It’s unclear if the paper fits to ICLR or should be submitted to a conference/journal that’s dedicated to the topic.
- Some parts of the paper are very technical and don’t contain information that is essential for a research paper.
- Weak baselines, lacking ablations, methods studied fail to reflect the real issue (the premise is that a standard de-identification approach failed, yet, the data studied is full of synthetic PII which is presumably clearly presented (data that failed to be de-identified may be less clearly recognizable data and be significantly harder to retrieve).

**Questions:**

- In what sense is the proposed method specific to medicine? If not specific, then why focus specifically on that topic?
- Figures 1 and 2 are unclear. Could you add a more detailed caption that explains what is shown?
- Is there evidence that de-identification failures/leakage are a common issue for such healthcare assistant system? What evidence would point to this? Can you demonstrate what happens with your synthetic PII if a de-identification method was employed?
- Who exactly is the assumed adversary in this instance? Are healthcare agents with access to large amounts of de-identified patient medical notes realistic, and, who would have access to such agents? I cannot imagine why anyone but a physician would need an LLM agent with RAG abilities of patient medical notes.

---

> ### Author Response · Authors · 2025-11-22
>
> We thank the reviewers for their consideration and comments. We answer their comments to the best of our ability in the following manner.
>
> **Comment 1**: The proposal reads like a stack of existing tools and ideas that are used to solve a real world problem rather than a novel method published at a scientific conference or journal.
>
> **Answer 1**: While our framework utilizes existing components, **our main contribution lies in the usage of medical signatures for semantic obfuscation**. Our primary goal is to highlight a potential exploitation of a vulnerability through a novel pipeline integrating the medical signature concept to provide additional obfuscation. Unlike existing full note attacks, which can be easily detected due to their full medical context, our pipeline helps mask them from the existing safety system in the LLM. To this extent, the primary novelty in our approach lies in the pipeline design and the integration of a medical signature, which balances medical obfuscation with sufficient context to retrieve accurate patient information without hallucinations. We add these concepts to Section 1 (Introduction).
>
> **Comment 2**: It’s unclear if the paper fits to ICLR or should be submitted to a conference/journal that’s dedicated to the topic.
>
> **Answer 2**: **Our submission aligns directly with the conference's focus on safety, privacy, and trustworthy machine learning**. Additionally, recent ICLR publications, such as [1], have established a precedent for empirical studies on PII leakage and model vulnerabilities. **We argue that identifying and characterizing these vulnerabilities is not a necessity for the responsible deployment of LLMs in critical environments**. The primary goal of this paper is to introduce a novel adversarial framework that systematically exploits a vulnerability in LLM security. This exploit is unaddressed by modern defenses.
>
> **Comment 3**: Some parts of the paper are very technical and don’t contain information that is essential for a research paper.
>
> **Answer 3**: We have included these comments for reproducibility but we have checked the text to make it clearer
>
> **Comment 4**: Weak baselines, lacking ablations, methods studied fail to reflect the real issue (the premise is that a standard de-identification approach failed, yet, the data studied is full of synthetic PII which is presumably clearly presented (data that failed to be de-identified may be less clearly recognizable data and be significantly harder to retrieve).
>
> **Answer 4**: While our work assumes scenarios where de-identification fails, we acknowledge that de-identification is a limiting technique in itself, reducing the scope of the healthcare model. In the event that de-identification is present but fails, our attack would still be effective in retrieving the data that remains in the dataset. We detail this description in Section 3 (Healthcare-LLM).
>
> **Comment 5**: In what sense is the proposed method specific to medicine? If not specific, then why focus specifically on that topic?
>
> **Answer 5**: Given the high dimensionality of medical terms and the large entropy within medical notes between different patients, medical signatures perform better in the medical field, as extractions tend to contain more unique terms. In other relevant fields, this is not always the case, as the amount of uniqueness significantly decreases. As a result, while our approach can work in those fields, it would be less effective. To address this, we craft a pipeline relevant to the medical context, where signatures have higher uniqueness.
>
> **Comment 6**: Figures 1 and 2 are unclear. Could you add a more detailed caption that explains what is shown?
>
> **Answer 6**: We have modified both figures 1 and 2 captions to be more specific.

---

> ### Author Response · Authors · 2025-11-22
>
> **Comment 7**:  Is there evidence that de-identification failures/leakage are a common issue for such healthcare assistant systems? What evidence would point to this? Can you demonstrate what happens with your synthetic PII if a de-identification method were employed?
>
> **Answer 7**:  We thank the reviewers for their consideration and expertise. Existing work has showcased that data de-identification is not necessarily precise. Works such as [2] and [3] both argue that data de-identification can often fail, as Named Entity Recognition techniques typically rely on models that often fail in complex names. In the case of our approach, if synthetic PII were de-identified after the fact, some of the information would be removed from the knowledge base. In truth, no leakage model would be capable of leaking the information, as the model would not know what to respond, resulting in hallucinations. Our approach would similarly reach the same relative percentage of leakage as before. In short, given that our approach leaked for most medical notes, it can be assumed that if NER failed a small number of times during de-identification, then our approach would also leak that small amount of data left over. Furthermore, we emphasize in our write-up that de-identification is a significant detriment to healthcare agent utility, which commonly needs access to some PII for tasks such as billing or patient retrieval.
>
> **Comment 8**:  Is there evidence that de-identification failures/leakage are a common issue for such healthcare assistant systems? What evidence would point to this? Can you demonstrate what happens with your synthetic PII if a de-identification method were employed?
>
> **Answer 8**: We wish to clarify that the attack model is not limited only to physicians but also to any actor with access to the medical assistant. This could include individuals from administrative departments, such as billing and medical coding, as well as external collaborators, including specialists. We assume the attacker has some form of internal access, either through a backdoor into the medical hospital network or as an insider attack. We also assume the adversary is not at the highest administrative level and can obtain full access to the RAG database or model internals; as a result, he will operate in a black box approach. We add these details to Section 4 (Threat Model).
>
> We want to highlight all changes to the paper can be seen in the updated manuscript.
>
> [1] Robin Staab, Mark Vero, Mislav Balunovic, and Martin Vechev. Beyond memorization: Violating privacy via inference with large language models. In The Twelfth International Conference on Learning Representations, 2024. URL https://openreview.net/forum?id=kmn0BhQk7p.
> [2] Nils Lukas, Ahmed Salem, Robert Sim, Shruti Tople, Lukas Wutschitz, and Santiago Zanella-Beguelin (2023). Analyzing leakage of personally identifiable information in language models. In 2023 IEEE Symposium on Security and Privacy (SP), pp. 346–363. IEEE, 2023.
> [3] Atiquer Rahman Sarkar, Yao-Shun Chuang, Noman Mohammed, and Xiaoqian Jiang (2024). De-identification is not always enough. arXiv preprint arXiv:2402.00179, 2024.

---

### Official Review · Reviewer_DFr3 · 2025-10-26

**Soundness:** 3
**Presentation:** 2
**Contribution:** 2
**Rating:** 2
**Confidence:** 4

**Summary:**

This paper proposes a attack on healthcare LLMs that uses retrieval augmented generation. The key innovation is extracting "medical signatures", which are combinations of unique medical terms in clinical notes, from de-identified medical notes to craft adversarial prompts that causes LLMs to leak Private Identifiable Information (PII). The authors claim their approach achieves 98% attack success rate, significantly outperforming existing methods (≤10%).

**Strengths:**

1. Healthcare AI privacy is critical given HIPAA regulations and increasing LLM deployment in medical settings. The paper makes contribution to an important problem timely.

2. The experiment results are impressive. Compared to previous attack techniques that have a leakage rate of up to 10%, this attack method achieves a leakage rate of up to 98%.

3. The method itself is an interesting variant of current common attacks that uses non-PII that is highly related to PII. Using condensed medical signatures is clever since it provides semantic obfuscation while maintaining enough context for accurate retrieval.

**Weaknesses:**

1. The idea of using non-PII that is highly related to PII to bypass safety guards and maliciously recover PII have been explored by many researcher [1-4]. Since many papers have presented evidence demonstrating that this kind of attacks is feasible and effective, the core contribution of this paper is limited. Uniqueness-based extraction of highly revealing medical non-PII may be the only contribution of this paper.

2. The paper identifies vulnerabilities but offers no mitigation strategies. This limits practical impact and may lead to actual harms.

3. The assumption that attackers possess de-identified versions of exact medical notes stored in the RAG database is quite strong. The paper does not adequately address cases where attackers only have partial or approximate medical information.

References
[1] Kandpal, N., Pillutla, K., Oprea, A., Kairouz, P., Choquette-Choo, C., & Xu, Z. (2024). User Inference Attacks on Large Language Models. In Proceedings of the 2024 Conference on Empirical Methods in Natural Language Processing, pp. 18238-18265.
[2] Powar, J., & Beresford, A. R. (2023). SoK: Managing Risks of Linkage Attacks on Data Privacy. Proceedings on Privacy Enhancing Technologies, 2023, 97-116.
[3] Anderson, M., Amit, G., & Goldsteen, A. (2024). Is My Data in Your Retrieval Database? Membership Inference Attacks Against Retrieval Augmented Generation. arXiv:2405.20446.
[4] Liu, R., Wang, T., Cao, Y., & Xiong, L. (2024). Precurious: How innocent pre-trained language models turn into privacy traps. In Proceedings of the 2024 ACM SIGSAC Conference on Computer and Communications Security, pp. 3511-3524.

**Questions:**

1. In real-world clinical deployments, hospitals and healthcare organizations often deidentify the knowledge base for RAG models as well for the sake of strict HIPAA compliance. In this context, is your attack still effective? If so, how?

2. Do you have suggestions for future work on how we can try to defend this kind of attacks? Providing novel defense strategies is as important as discovering novel attacks. I understand the paper's scope does not include providing a defense strategy. But, it is researchers' due diligence to briefly talk about potential future work in the conclusion. In this case, a critical future work would be the defense strategy for such an attack.

3. In Algorithm 2, the choice of A=20 for early stopping appears arbitrary. Could you please justify this choice?

4. Existing attack methods perform significantly poorly. Are baselines optimally implemented? Why do they fail so dramatically?

**Details Of Ethics Concerns:**

The paper presents a adversarial attack that can be used against real-world LLM systems deployed in clinical scenarios without providing its corresponding defense strategy. Malicious users can easily use this attack to obtain PII information. Such privacy breach is concerning.

---

> ### Author Response · Authors · 2025-11-22
>
> We thank the reviewer for their consideration and expertise, as well as for their comments on our work.
>
> **Comment 1**: The idea of using non-PII that is highly related to PII to bypass safety guards and maliciously recover PII have been explored by many researcher [1-4]. Since many papers have presented evidence demonstrating that this kind of attack is feasible and effective, the core contribution of this paper is limited. Uniqueness-based extraction of highly revealing medical non-PII may be the only contribution of this paper.
>
> **Answer 1** While papers [1-4] establish a foundation for LLM data leakage, our research occupies a distinct niche within its broader domain. Works such as [1] and [3] primarily focus on knowledge reconstruction from partially known information; in other words, they attempt to identify samples in the knowledge base from some known examples. Work [1] explores this aspect for pretrained models, while [3] applies it to an RAG interface. Work [2] provides an overview of the state of knowledge in the area of model leakage, including targeted attacks and training data inference, but does not detail any improvements or effective comparisons of the attacks. Work [4] presents a distinct attack approach, focusing on instilling a training-data poisoning attack to encourage open-source models to leak data more readily. **In the scope of the existing work, our implementation introduces a novel pipeline for a more optimized attack vector in RAG PII leakage. Our framework offers three distinct key differences from these existing works: its application to healthcare, the use of medical signatures, and a novel attack pipeline**. We add these details to Section 2 (Related Works).
>
> **Comment 2**: 2.	The paper identifies vulnerabilities but offers no mitigation strategies. This limits practical impact and may lead to actual harm.
>
> **Answer 2**: While the primary scope of this work is to demonstrate a novel attack pipeline exploiting vulnerability in healthcare RAG Agents, we agree to provide potential defense to mitigate harm. To address this, we propose a “note reconstruction” defense that utilizes a secondary model and RAG architecture to reconstruct the medical note. After the medical notes are reconstructed, they will be used to query the model. In benign tasks, this will return correctly; otherwise, it will remove the obfuscation and cause it to fail like the baselines. We have added the details of the mitigation to our conclusion.
>
> **Comment 3**: The assumption that attackers possess de-identified versions of exact medical notes stored in the RAG databases is quite strong. The paper does not adequately address cases where attackers only have partial or approximate medical information.
>
> **Answer 3**: The assumption that an attacker possesses full de-identified medical notes is consistent with established literature such as Lukas et al. [5]. In most papers, leveraging full medical notes for de-identification access to the de-identified notes is considered. In future work to evaluate the robustness of our approach, we propose examining how our attack functions with partial medical notes. We add these details to Section 4 (Threat Model).
>
> **Comment 4**: In real-world clinical deployments, hospitals and healthcare organizations often deidentify the knowledge base for RAG models as well for the sake of strict HIPAA compliance. In this context, is your attack still effective? If so, how?
>
> **Answer 4**: We highlight that de-identification is a restrictive defense that limits the model’s operational scope for medical assistant tasks in areas like billing and patient retrieval. Therefore, the scope of this work explores the area where de-identification is not leveraged, as the reduced scope of work significantly limits the model's capabilities. Additionally, in the case of our attack, it will be able to retrieve data that is still identifiable due to misses if the dataset was de-identified using a de-identification technique. We describe these details and tradeoffs regarding de-identification in Section 3.
>
> **Comment 5**: Do you have suggestions for future work on how we can try to defend this kind of attacks? Providing novel defense strategies is as important as discovering novel attacks. I understand the paper's scope does not include providing a defense strategy. But, it is researchers' due diligence to briefly talk about potential future work in the conclusion. In this case, a critical future work would be the defense strategy for such an attack.
>
> **Answer 5**: We have taken this consideration into mind and have previously answered this in comment 2. Previously, we proposed a specific mitigation strategy leveraging note rebuilding. We present our solution in Section 8 (Conclusion).

---

> > ### Author Response · Authors · 2025-11-22
> >
> > **Comment 6**: In Algorithm 2, the choice of A=20 for early stopping appears arbitrary. Could you please justify this choice?
> >
> > **Answer 6**: The value A=20 was not arbitrarily chosen for our experiments; rather, in this case, it corresponds to the maximum number of adversarial templates used. For this experiment, we aimed to assess the speed at which our approach caused model leakage across all templates. We show that setting A to a high value is not detrimental, as most leakage success occurs in a few attempts. More practical cases can reduce A to a value slightly larger than five to remove outliers, as most cases will already leak by then. We describe this in Section 6 (Implementation).
> >
> > **Comment 7**: Existing attack methods perform significantly poorly. Are baselines optimally implemented? Why do they fail so dramatically?
> >
> > **Answer 7**: To the best of our knowledge, we implemented the baselines as well as we could while aligning to our primary case (RAG interface). Techniques not primarily designed for RAG agents, as a result, had to be modified, also in accordance with our attack model. The primary reason for their poor performance is the lack of message obfuscation, as the entire medical note is provided as context. As a result, the context remains explicit and is easily recognized by the model. The model will therefore reject the request as specified in its fine-tuning and its safety prompt. We highlight the reason for the results in Section 7 (Evaluation).
> >
> > Changes to the proposed sections, as well as general revisions for clarity, are reflected in the updated manuscript.
> >
> > [1] Kandpal, N., Pillutla, K., Oprea, A., Kairouz, P., Choquette-Choo, C., & Xu, Z. (2024). User Inference Attacks on Large Language Models. In Proceedings of the 2024 Conference on Empirical Methods in Natural Language Processing, pp. 18238-18265.
> > [2] Powar, J., & Beresford, A. R. (2023). SoK: Managing Risks of Linkage Attacks on Data Privacy. Proceedings on Privacy Enhancing Technologies, 2023, 97-116.
> > [3] Anderson, M., Amit, G., & Goldsteen, A. (2024). Is My Data in Your Retrieval Database? Membership Inference Attacks Against Retrieval Augmented Generation. arXiv:2405.20446.
> > [4] Liu, R., Wang, T., Cao, Y., & Xiong, L. (2024). Precurious: How innocent pre-trained language models turn into privacy traps. In Proceedings of the 2024 ACM SIGSAC Conference on Computer and Communications Security, pp. 3511-3524.

---

> ### Comment · Reviewer_DFr3 · 2025-11-26
>
> I thank the authors for the rebuttal.
> The fact that your work occupies a niche does not contradict with my claim that your contribution is limited.
> As you have acknowledged, the three differences from the existing works are its application to healthcare, the use of medical signatures, and a novel attack pipeline. Applying to healthcare does not count as a contribution. I have acknowledged the other two contributions in the last sentence of my Comment 1. Thus, I keep my claim that your contribution is limited.
> I appreciate that the authors tried to propose a mitigation method. However, it is just a few sentences in the conclusion describing the proposal. A "future work" section without experiments providing evidence to support the proposed method is still not sufficient. Thus, my claim in Comment 2 remains unaffected.
> My concerns in Comments 3, 4, 6, and 7 are addressed by the authors.
> Considering that all my minor concerns are addressed and my two major concerns are not addressed, I raise the rating from 2 to 4 and change the confidence from 4 to 5. Unless the authors can expand the scope and add more experiments, my rating is not likely to change again.

---

### Official Review · Reviewer_idTn · 2025-11-01

**Soundness:** 2
**Presentation:** 3
**Contribution:** 2
**Rating:** 6
**Confidence:** 4

**Summary:**

This paper introduces a framework for extracting unique medical signatures from de-identified medical notes and building adversarial prompts using those signatures. The authors implement and test their framework using a publicly known medical notes dataset (MT-Samples) with augmented artificial PII data. They compare their attack framework against existing approaches and achieve up to 98% leakage rate. The paper is well written, and the problem is a pressing one.

**Strengths:**

1. Figures 1-3 and Algorithms 1-2 make it clear what the authors are doing.
2. The authors achieve high leakage rates with their attack approach, which indicates that this problem is urgent.
3. The proposed attack framework incoporates many simple but well thought-out mechanisms. One could see how this framework could be extended to further increase the empirical leakage rates (which further increases the sense of urgency for this problem).

**Weaknesses:**

1. The authors could have tried out more than the two safeguards that they used.
2. The empirical study uses a publicly known medical notes dataset (MT-Samples). It would be interesting to see how much leakage occurs on a private dataset.

**Questions:**

1. How much does the amount of leakage depend on the specific de-identification techniques used? How much would a more robust de-identification technique decrease leakage?
2. What other safeguards could be implemented to mitigate leakage from your signature-based attack?

---

> ### Author Response · Authors · 2025-11-22
>
> We thank the reviewer for their consideration and expertise and for their comments on our work.
>
> **Comment 1**: The authors could have tried out more than the two safeguards that they used.
>
> **Answer 1**: We acknowledge the observation that only two defensive strategies were utilized to mitigate the threats to the LLM; however, we selected these specific strategies because they offer the least operational intrusion. System prompts are standard methods for ensuring secure performance, while LLM safety fine-tuning is a universal practice that utilizes reinforcement learning with human feedback. In future work, we intend to examine additional safety mechanisms that reduce LLM capabilities, such as Named Entity Recognition (NER) and external prompt semantic checkers like Llama Guard. We have updated Section 8 (Conclusion) to reflect this focus.
>
> **Comment 2**: The empirical study uses a publicly known medical notes dataset (MT-Samples). It would be interesting to see how much leakage occurs on a private dataset.
>
> **Answer 2**: We acknowledge that testing proprietary data would provide a valuable perspective on leakage. However, due to the strict privacy regulations and ethical concerns in medical data acquisition, acquiring high-quality, large-scale private data provides a significant challenge and reduces reproducibility. As a result, we utilized MTSamples, a widely established benchmark in medical NLP, which comprises real, de-identified medical transcripts across various fields. To address the reviewer’s comment, we have clarified in our paper that, for future work, we intend to explore a small private clinical environment. We have updated Section 6 (Implementation) accordingly.
>
> **Comment 3**: How much does the amount of leakage depend on the specific de-identification techniques used? How much would a more robust de-identification technique decrease leakage?
>
> **Answer 3**: We specify that, in our framework, model leakage is defined as the extraction of PII from the model's knowledge through a security vulnerability or bypass. What de-identification does is directly remove the information from the model's knowledge, leaving nothing to extract or leak. In the case of effective attacks, the de-identified data would still leak, but the number would be reduced as the total knowledge of the model is removed. While robust de-identification reduces the impact of model leakage, removing PII simultaneously degrades model performance. Medical assistants for tasks such as patient information retrieval, billing, or note assistance typically require access to patient PII information; removing them significantly reduces the knowledge base of the model and impedes it from performing the expected function. We added these details to Section 3.
>
> **Comment 4**: What other safeguards could be implemented to mitigate leakage from your signature-based attack?
>
> **Answer 4**: Beyond the safeguards evaluated in, we propose two additional safeguards, namely NER, and external safety checkers such as Llama Guard. However, both safeguards are intrusive and can negatively impact system performance. NER-driven de-identification limits model access to restricted data, thereby restricting the model's capabilities. At the same time, Llama Guard, due to its external nature, can lead to over-filtering. We have expanded Section 3 to detail the impact and limitations of these safeguards.
>
> All according changes to the manuscript can be seen in the updated rebuttal.

---

### Meta-Review · Area_Chair_tD8A · 2026-01-09

**Summary:**

This paper studies privacy leakage risks in RAG-based medical assistants and investigates how adversarial prompting combined with “medical signatures” extracted from patient notes can be used to induce models to retrieve personally identifiable information (PII) from secured medical databases. The authors construct a RAG-based healthcare agent and demonstrate that signature-guided attacks can bypass existing safety mechanisms more effectively than prior adversarial prompting approaches. They report high leakage rates under their attack setting, suggesting vulnerabilities in current RAG-based medical systems.

Reviewers raised concerns that the contribution offers limited novelty beyond applying existing techniques to the medical domain, and that the evaluation is weak, with limited baselines and experiments. While the rebuttal partially addressed these issues, key concerns remain unresolved. Therefore, we recommend rejecting this work.

**Reviewer Concerns:**

For Reviewer idTn, the two detailed questions regarding (1) the dependence of leakage on de-identification techniques and (2) potential safeguards have been adequately addressed. However, the main concerns—that the authors evaluated only two safeguards and that the experiments are limited to a public dataset (MT-Samples), with no evaluation on private or more realistic data—remain deferred to future work.

For Reviewer DFr3, the authors’ rebuttal has adequately addressed several of the technical and clarifying concerns (specifically those raised in Comments 3, 4, 6, and 7). The reviewer also acknowledges the introduction of medical signatures and a novel attack pipeline as contributions. However, the reviewer maintains that the overall contribution remains limited. Applying existing attack methodologies to the healthcare domain alone does not constitute a substantial research contribution, and the proposed mitigation strategy is only briefly discussed without empirical validation. As such, the major concerns regarding limited novelty and lack of experimental support for mitigation remain unresolved. As a result, while the reviewer has increased their score from 2 to 4 and their confidence from 4 to 5 in recognition of the clarifications provided.

For Reviewer yn4e, the rebuttal did not change their assessment that the paper reads primarily as a stack of existing tools rather than a novel machine learning contribution, and that the experimental design does not convincingly support the claims due to limited scope, weak baselines, and unrealistic assumptions about de-identification failures.

**Reviewer Scores:**

Reviewer DFr3 has changed the score from 2 to 4 after the discussion.

Reviewer idTn would not change the score since the major weakness pointed out by this reviewer do not get addressed.

Reviewer yn4e might increase the score (reject) but it might still not be positive due to the major concerns unaddressed.

---

### Decision · Program_Chairs · 2026-01-26

Reject